# Experimental Assessment of Zika Virus Mechanical Transmission by *Aedes aegypti*

**DOI:** 10.3390/v11080695

**Published:** 2019-07-31

**Authors:** Antoine Boullis, Nadège Cordel, Cécile Herrmann-Storck, Anubis Vega-Rúa

**Affiliations:** 1Laboratory of Vector Control Research, Institut Pasteur of Guadeloupe—Lieu-dit Morne Jolivière, 97139 Les Abymes, Guadeloupe, France; 2Dermatology and Clinical Immunology Unit, Guadeloupe University Hospital, 97159 Pointe-à-Pitre Cedex, Guadeloupe, France; 3INSERM Unite 1234, Normandie University, UNIROUEN, 76000 Rouen, France; 4Microbiology Laboratory, Guadeloupe University Hospital, 97159 Pointe-à-Pitre Cedex, Guadeloupe, France

**Keywords:** ZIKV, *Aedes aegypti*, Arbovirus, Mechanical transmission

## Abstract

The pandemic emergence of several mosquito-borne viruses highlights the need to understand the different ways in which they can be transmitted by vectors to human hosts. In this study, we evaluated the propensity of *Aedes aegypti* to transmit mechanically Zika virus (ZIKV) using an experimental design. Mosquitoes were allowed to feed on ZIKV-infected blood and were then rapidly transferred to feed on ZIKV-free blood until they finished their meal. The uninfected blood meals, the mosquito abdomens, as well as the mouthparts dissected from fully and partially engorged mosquitoes were analyzed using RT-qPCR and/or virus titration. All the fully engorged mosquito abdomens were ZIKV-infected, whereas their mouthparts were all ZIKV-negative. Nonetheless, one of the partially engorged mosquitoes carried infectious particles on mouthparts. No infectious virus was found in the receiver blood meals, while viral RNA was detected in 9% of the samples (2/22). Thus, mechanical transmission of ZIKV may sporadically occur via *Ae. aegypti* bite. However, as the number of virions detected on mouthparts (2 particles) is not sufficient to induce infection in a naïve host, our results indicate that mechanical transmission does not impact ZIKV epidemiology.

## 1. Introduction

Recent outbreaks of emerging mosquito-borne arboviruses highlight the need to understand the transmission dynamics of these diseases. Zika virus (ZIKV), a *Flaviviridae* from the genus *Flavivirus*, caused recent pandemics throughout the world, from the Micronesia region in 2007 to the Americas and the Caribbean in 2016 [1]. Like many other types of *Flaviviridae*, ZIKV relies predominantly on biological transmission to spread in human populations: it needs to be ingested by a competent vector to multiply and to be transmitted to a susceptible new host. In this prospect, *Aedes* mosquitoes are the major insect vectors for this virus. Nevertheless, other transmission routes may affect the dynamics of mosquito-borne diseases like ZIKV. Recent research evidenced “within-mosquitoes” transmission of ZIKV in two vector species (*Ae. aegypti* & *Ae. albopictus*), with both venereal and vertical transmission mechanisms (Reviewed in reference [1]). On another hand, human-to-human transmission of ZIKV has been also reported [2] via sexual, intrapartum, and intrauterine transmissions.

Mechanical transmission is a mechanism described for several arthropod vectors consisting of a simple transfer of pathogens via the mouthparts of a hematophagous insect from an infected host to a susceptible one [3]. Such mechanical transfer must occur in a short period of time between two feeding events. *Ae. aegypti* mosquitoes have been shown to mechanically transmit lumpy skin disease virus [4] and even pathogens that predominantly rely on biological transmission such as the chikungunya virus [5]. However, as the biological transmission constitutes the primary means of transfer of arbovirus by mosquitoes, the mechanical transmission is often neglected and evidences are still sparse regarding this subject, especially for mosquito-transmitted *Flaviviruses* [3]. Thus, this phenomenon needs to be deepened as recommended in literature [6]. As mechanical transmission of ZIKV can influence the epidemiology of this virus, in this paper we evaluated the propensity of *Ae. aegypti* mosquitoes to transmit mechanically ZIKV under laboratory conditions using a comprehensive experimental design.

## 2. Material & Methods

### 2.1. Ethics Statement

This study has been approved by the internal ethics committee of the Pasteur Institute of Guadeloupe (established since September 2015). There is no agreement number for internal ethics board. Anubis Vega-Rúa and Antoine Boullis (authors of the study) provided written consent for blood donation to artificially feed mosquitoes in experiments.

### 2.2. Insect Rearing

Hundreds of *Ae. aegypti* eggs (Rockefeller colonized strain) were placed in dechlorinated tap water until hatching. Larvae were fed with rabbit pellets until pupation. Water and food were renewed every 2–3 days. Emerging adults were kept in flight cages under controlled conditions (27 ± 1 °C; 70 ± 10% RH; 12:12 h L:D photoperiod) and fed with 10% sucrose solution. Ten to 15 days old female mosquitoes were starved for 24 h before their use in experiments.

### 2.3. Virus Strain

A ZIKV strain isolated from *Ae. taylori* in Senegal was used in the experiments (GenBank accession number: KU955592) [7]. A third passage (produced in BHK-21 cell line) was provided as lyophilisate by the Emergence Virus Unit (Marseille) via the initiative “European Virus Archive goes global” (EVAg) and re-suspended into DMEM (Gibco^®^, Life Technologies^TM^, Paisley, UK) for viral production in our laboratory. Vero cells (ATCC, ref. CCL-81) were used for virus culture, provided with DMEM supplemented with 2% Foetal Bovine Serum (FBS–Eurobio, Les Ulis, France), with a multiplicity of infection of 0.1. Supernatants were harvested 3 days later and stored at −80 °C before their use in experiments. The viral titer of the stock (P4) was estimated using a 10-fold serial dilutions on Vero Cells (ATCC, ref. CCL-81), and expressed in Tissue Culture Infectious Dose 50 (TCID_50_) per mL.

### 2.4. Artificial Infection Procedure

Mosquito feeding was performed using a Hemotek system (Hemotek Ltd.^®^, Blackburn, UK). Starved females were individually placed into acrylic tubes (WHO standard insecticide bioassay tubes), closed on both sides by clean nets. One Hemotek reservoir (called a *source capsule*) was filled with 1.4 mL of washed human erythrocytes and 700 µL of ZIKV suspension supplemented with the phagostimulant adenosine triphosphate (ATP; Sigma-Aldrich^©^, St. Louis, MO, USA) at a final concentration of 5 mM. The blood meal titer in the *source capsule* was 7 log_10_ TCID_50_/mL and verified after the experiment via TCID_50_ assays. The *receiver capsules* were each filled with 2 mL of uninfected washed blood supplemented with ATP at 5 mM. The reservoirs were sealed with clean pork intestine as a feeding membrane.

As soon as the mosquito was transferred to the *receiver capsule* (Figure 1B,C), a new mosquito was placed to feed on the *source capsule* (Figure 1A). The feeding status of mosquitoes placed on the *receiver capsule* was checked every ten minutes. The replicate was considered to be completed when the female was fully engorged. If the female was not further engorged 30 min after the transfer, the tube was discarded (to limit the degradation on viral particles eventually located on the mouthparts). When mosquito females were fully engorged, the corresponding reservoir was immediately removed from the Hemotek feeder and the blood, the feeding membrane and the mosquito were separately stored at −80 °C until further analyses. Another *receiver capsule* was then placed on the Hemotek feeder to conduct a new experimental replicate. The experiment was limited to 60 min to avoid any significant decrease in the infected blood meal titer. A total of 22 *receiver capsules* were collected for further analyses.

In parallel, 25 female mosquitoes (called partially engorged females) were instantly frozen and killed after feeding on the *source capsule* to check if some viral particles were present in the mouthparts of mosquitoes after incomplete blood feeding. Their proboscises were immediately analyzed as described in the next section.

### 2.5. Sample Analysis

Samples were used for ZIKV screening after the experimental infection (i.e., blood meals, feeding membranes, fully and partially engorged mosquitoes) by using different techniques.

#### 2.5.1. Blood Samples

Blood samples (Figure 1D) were filtered using Minisart 0.22 µm sterile filters (Sartorius AG, Goettingen, Germany). One third (≈ 150 µL) of each filtered blood sample was diluted into 500 µL of DMEM supplemented with 2% FBS and inoculated onto Vero cells (ATCC, ref. CCL-81) cultures in 25 cm^2^ flasks for viral isolation through a method described by Vazeille and collaborators [8]. Supernatants were harvested after 6 days for titration and monolayers were fixed with a solution of 10% formalin, 0.2% crystal violet and 20% ethanol to reveal an eventual cytopathogenic effect (CPE). Supernatants were then titrated using plaque assays (6-well plate), using a method described by Arias-Goeta and colleagues [9] to confirm the infectious status of the samples. Another third of each filtered blood samples was directly titrated by a TCID_50_ assay (96-well plate), using a method described by Hery and colleagues [10]. Finally, the last third of each blood sample was used for RNA extraction and Reverse Transcription coupled with quantitative Polymerase Chain Reaction (RT-qPCR). RNA extractions were performed with the Nucleospin RNA kit (Macherey-Nagel GmbH, Düren, Germany) following the manufacturer’s instructions. Amplification of ZIKV specific RNA was done by real time RT-qPCR (Applied Biosystem^®^ 7500, Foster City, USA) using primers described in Lanciotti and collaborators [11] (1086–1162c) and a *Power* SYBR^®^ Green RNA-to-C_T_^TM^
*1-step* kit (Applied Biosystems^®^) according to the supplier’s instructions. The thermal profile used was the following: 30 min of reverse transcription at 50 °C, amplification at 95 °C for 2 min, 45 cycles at 95 °C for 15 s followed by 60 °C for 1 min, and finally, a melt curve stage. To estimate the quantity of viral RNA (vRNA) in the positive samples, a linear equation was built using ten-fold serial dilutions of standards (Real Star^®^ Zika virus RT-PCR kit; Altona GmbH, Hamburg, Germany) with known amount of vRNA copies. The linear equation “y = −0.3333x + 12.5” was obtained, where “y” corresponds to the log_10_ of vRNA copies and “x” to the corresponding C_T_.

#### 2.5.2. Feeding Membranes

Feeding membranes (Figure 1D) were individually placed in a 15 mL Falcon conical tube with 2 mL of DMEM + 2% FBS + 1X Anti-Anti (Gibco^®^) for a 30 min elution followed by a 5 min centrifugation to recover a maximum of eluate. The half of each sample (1 mL) was inoculated onto Vero cells (ATCC, ref. CCL-81) cultures for viral isolation following the same protocol described for filtered blood, while 500 µL were directly titrated by plaque assay (6-well plate) as described above. In all cases, monolayers were fixed after the incubation period as described above to reveal the eventual CPE. The rest of the eluate (200 to 400 µL) was used for RNA extraction and RT-qPCR assays as described for blood samples.

#### 2.5.3. Mosquitoes

Proboscises and abdomens of fully engorged mosquitoes were dissected (Figure 1E), as well as the proboscises of partially engorged mosquitoes that only fed on the *source capsule* (Figure 1F). All the samples were tested individually, except the proboscises of fully engorged mosquitoes that were pooled (up to 4 samples per pool), and were crushed with glass beads in 300 µL of DMEM + 2% FBS + 1X Anti-Anti. All the samples were titrated by plaque assays (6-well plates) as described above.

## 3. Results & Discussion

To determine the amount of virus that mosquitoes ingested during the interrupted infectious blood meal at 7 log_10_ TCID_50_/mL, the abdomens of mosquitoes were screened for ZIKV infection after feeding on the *receiver capsules*. All the 22 mosquito abdomens were ZIKV-positive with a mean viral load estimated at 1.88 ± 1.31 log_10_ PFU/abdomen (Mean ± S.E.) (Table 1). These results confirm that mosquitoes acquired enough blood from the *source capsule* to ingest infectious viral particles, and to eventually carry some of these viral particles on their mouthparts. However, the examination of the corresponding mouthparts did not reveal any infectious viral particle in any of the pools analyzed. Two main hypotheses may explain these results: either mosquitoes carried some viral particles and released them during their second blood intake (on the *receiver capsule*), or they did not carry any viral particle on their mouthparts at all.

In accordance with these hypotheses, it is important to know if mosquitoes can carry some viral particles on their mouthparts just after the intake of an infectious blood meal. The analysis of partially engorged mosquitoes that fed only on the infected *source capsule* revealed that one out of the 25 tested individuals carried two infectious ZIKV particles on the proboscis (Table 1). Even if the infection rate of proboscises was very low (4%), this result demonstrates for the first time that *Ae. aegypti* can carry infectious ZIKV on its mouthparts. In parallel, the results obtained from analyses achieved on the *receiver capsule* membranes were quite consistent with these findings, because even if no infectious ZIKV particles were detected, ZIKV vRNA was found in two samples (Table 1; Figure 2). However, the high C_T_s obtained for both membrane samples (i.e., 37.9 and 41.9) did not allow absolute quantification of vRNA using the linear equation described above as they are under the quantification threshold. We thus estimated that the amount of vRNA copies is less than 500 vRNA copies/sample. These results are consistent with the number of ZIKV particles found in partially engorged mosquitoes mouthparts. Finally, we did not reveal the presence of ZIKV infectious particles nor vRNA in blood samples (Table 1). These findings mean that ZIKV material can be mechanically transmitted from the *source capsule* to any of the *receiver capsules*. In addition, they highlight the usefulness of artificial feeding devices on mechanical transmission studies, especially when highly-dosed source meals are used. Artificial feeding devices similar to those of the present study have successfully been employed to evidence the mechanical transmission of *Besnoitia besnoiti* genetic material by the hematophagous fly *Stomoxys calcitrans* from an infected artificial feeder to a pathogen-free one [12].

For these analyses, distinct methodologies were used to obtain different information regarding the viral infection. Indeed, titration using plaque assays is a sensitive approach allowing detection of few infectious viral particles [13], while vRNA isolation and amplification allows sensitive detection of vRNA in samples where the virus lost its infectiousness. In addition, the entire sample volume was used for viral detections with both approaches to increase the probabilities of viral particles and/or genomes detection in the sample.

The feeding behavior of *Ae. aegypti* and its anthropophilic habits [14] are favorable arguments for mechanical transmission of pathogens to human hosts, as this mosquito can bite different people in a short period of time, allowing a rapid mechanical transfer of environmentally labile pathogens between two feeding events. However, in addition to mosquito feeding behavior, the success of mechanical transmission depends on the virus quantity on the vector mouthparts, and thus indirectly on both the virus titer of the blood source and the quantity of blood carried on mouthparts. These two factors are not favorable for ZIKV mechanical transmission by *Ae. aegypti*, because the amount of blood on mouthparts of a solenophagous dipteran does not exceed 10^−3^ nL after feeding [3] and the virus titer in the serum of a ZIKV viremic host is generally lower [15] than that used in the blood meal source in the present study (7 log_10_ TCID_50_/mL). Nevertheless, recent studies showed that the viral load in the skin of infected patients can significantly exceed the amount of virus in the blood [16]. This evidence suggests that high skin virus titer might favor ZIKV mechanical transmission. To test this hypothesis, the use of appropriate ZIKV-immunodeficient murine models such as *Ifnar1*^-/-^ mice [17] would be crucial to know if the host skin favors the mechanical transmission of ZIKV by *Ae. aegypti*.

Taken together, our results suggest that mechanical transmission of ZIKV from a viremic host by *Ae. aegypti* is rare. Firstly, it is a rare phenomenon for *Ae. aegypti* to carry viral infectious particles on its mouthparts. Secondly, even with a blood meal titer higher than the recorded human viremia, the number of infectious particles found in mosquito mouthparts (i.e., 2 virions) is insufficient to infect a naïve host. Indeed, studies conducted with murine models revealed that a dose of at least 10^2^ PFU ZIKV infectious particles is required to induce infection [17]. We can thus conclude based on our experimental setup, that even if *Ae. aegypti* is able to sporadically carry ZIKV particles in its mouthpart after feeding on infectious blood, the mechanical transmission of ZIKV by *Ae. aegypti* will not contribute to the spread of the infection to a new host and will not influence ZIKV epidemiology.

## Figures and Tables

**Figure 1 viruses-11-00695-f001:**
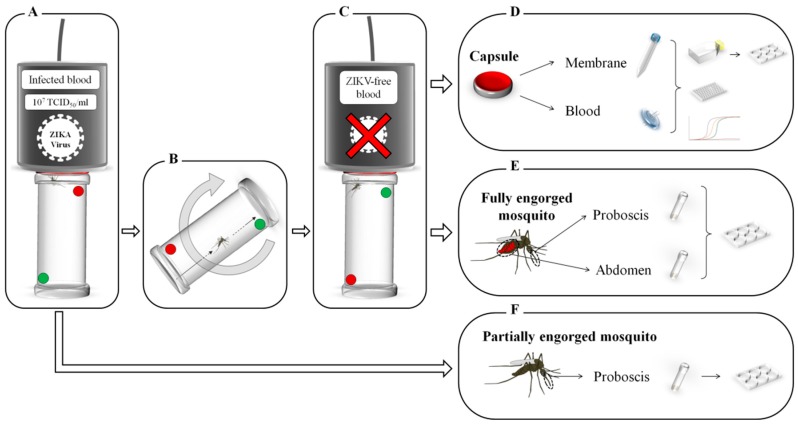
Experimental design used for the assessment of Zika virus (ZIKV) mechanical transmission by *Aedes aegypti*. (**A**) Mosquitoes were individualized in plastic tubes and allowed to start feeding on ZIKV-infected blood meal (called *source capsule*) until blood was visually detected in their abdomen. (**B**) Once blood was detected, the tube was turned upside down and transferred to a virus-free blood meal (called *receiver capsule*). (**C**) The mosquito was allowed to complete the blood meal for 30 min. (**D**) After engorgement, the *receiver capsule* compartments (feeding membrane and blood; *N* = 22) were screened for ZIKV by titration (virus culture, TCID_50_ and plaque assays) and by real time RT-qPCR. (**E,F**) Fully engorged mosquito compartments (abdomens and proboscises; *N* = 22) as well as proboscises of partially engorged mosquitoes (only fed on the source capsule; *N* = 25) were investigated for the presence of ZIKV using plaque assay titration.

**Figure 2 viruses-11-00695-f002:**
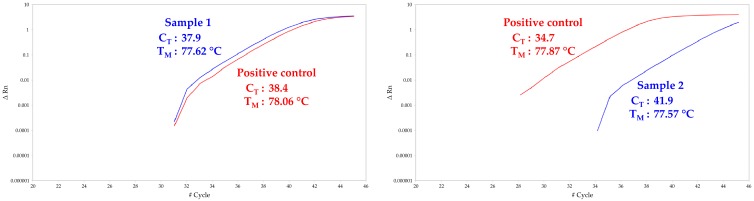
RT-qPCR amplification plots from the two positive feeding membrane samples. In blue, the amplification curves of each sample; in red, the amplification curves of positive control standards.

**Table 1 viruses-11-00695-t001:** Summary of the results from the different sample analyses.

Sample		N	ZIKV Positive Samples ^a^				Mean Titer (± S.E.) ^b^
			Titration		RT-PCR		
Source capsule	Blood	2	+	(2)	+	(2)	10^7^ ± 10^6.09^
Receiver capsule	Blood	22	–	(0)	–	(0)	NA
	Membrane	22	–	(0)	+	(2)	NA
Fully engorged mosquito	Mouthparts	22	–	(0)	NA		NA
	Abdomen	22	+	(22)	–	(0)	10^1.88^ ± 10^1.31^
Partially engorged mosquito	Mouthparts	25	+	(1)	NA		2

^a^ numbers in parentheses indicate the number of positive samples. ^b^ virus titers are expressed in TCID_50_/mL for blood samples, and in PFU/compartment for the rest of samples. ^c^ up to 4 samples were pooled.

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
