# Peer review of "Experimental Assessment of Zika Virus Mechanical Transmission by Aedes aegypti"

_viruses, 2019, doi:10.3390/v11080695_

Round 1

Reviewer 1 Report

In this article, A Boullis et al evaluate the effectiveness of mechanical transmission of Zika virus by aedes aegypti.

The introduction and methods are nicely writen and the objective is easy to understand as well as the obtained result that are sustained by multiple measurement of the viral load in the various compartment (blood, mosquitoes bodies and section etc..)..

My main question relay about the efficiency of the technical device used to induce a mechanical transmission. A more positive assay in which a clearly and loud transmission event will be evidenced with such an artificial infection procedure should be provided or discussed. I feel that this article and obtained result deserved such a positive demonstration.

Netherveless the current discussion that is limited to the result is of quality even if I'm not sure that murine models are good model to use. As most of immune competant mice are resistant to ZIKV infection, I suggest to clarify by indicating "susceptible mice model" or directly by naming the mice strain (IFNAR or others...)

I have a few minor comments:

in the description of the artificial infection procedure it is not clear if the blood in the receiver capsule is also supplemented with adenosine triphosphate.

Author Response

Response to Reviewers

Title: Experimental assessment of Zika virus mechanical transmission by Aedes aegypti

Journal: Viruses

On behalf of all co-authors, I thank the reviewers for their constructive remarks that contributed to improve the quality of our manuscript.

The manuscript was amended following the reviewer’s instructions, with changes highlighted in yellow. In this document, our answers to the reviewers’ comments are highlighted in bold. Line numbers referenced in our comments correspond to the revised version of the manuscript.

.

Anubis VEGA-RUA (Corresponding author)

Reviewer: 1

Comments and Suggestions for Authors

In this article, A Boullis et al evaluate the effectiveness of mechanical transmission of Zika virus by Aedes aegypti.

The introduction and methods are nicely writen and the objective is easy to understand as well as the obtained result that are sustained by multiple measurement of the viral load in the various compartment (blood, mosquitoes bodies and section etc..).

My main question relay about the efficiency of the technical device used to induce a mechanical transmission. A more positive assay in which a clearly and loud transmission event will be evidenced with such an artificial infection procedure should be provided or discussed. I feel that this article and obtained result deserved such a positive demonstration.

As explained in the manuscript, it might be difficult to accomplish a strong and loud mechanical transmission event with solenophagous dipterans, because they only carry few amounts of infected blood on their mouthparts.

One possibility to validate our experimental design (i.e. detection of viral particles in the receiver capsule), would be to test the device with another insect group, such as telmophagous dipterans, for which mechanical transmission events are well described (especially with horse flies). Experimental design with a similar artificial device has already been successfully used to assess and prove the mechanical transmission of Besnoitia besnoiti by the fly Stomoxys calcitrans (Liénard et al. 2013, Parasitol Res, 112: 479-486). According to the reviewer suggestion, we decided to add this reference in the manuscript as well as a short discussion regarding the efficiency of our device (lines 187 - 192).

In addition, we routinely use this device with the same parameters (blood meal titer, artificial membrane, fresh blood) to biologically infect mosquitoes with arboviruses, including Zika virus. The results that we obtain are very good (Chouin-Carneiro et al. 2016, PLoS NTD, 13: e0004543; Hery et al. 2019, Emerg Microbe Infect, 8: 699-706), proving the efficiency of artificial feeding device to infect mosquitoes.

Netherveless the current discussion that is limited to the result is of quality even if I'm not sure that murine models are good model to use. As most of immune competant mice are resistant to ZIKV infection, I suggest to clarify by indicating "susceptible mice model" or directly by naming the mice strain (IFNAR or others...).

We discussed about “murine model” in the manuscript because we think that living skin tissue may influence mechanical transmission efficiency, as it has been demonstrated that biological transmission is affected by the feeding source (artificial feeder vs. living animal). Furthermore, ZIKV replicates very well in skin tissues (Hamel et al. 2015, J Virol, 89: 8880-8896) and can even reach higher levels when compared to blood (Cordel et al. 2017, Br J Dermatol, 178: 108-110).

As requested by the reviewer, we reworded the sentence (lines 211- 213) as follows: “To test this hypothesis, the use of appropriate ZIKV-immunodeficient murine models such as Ifnar1-/- mice (Lazear et al. 2016) would be crucial to know if the host skin favors the mechanical transmission of ZIKV by Ae. aegypti.”

I have a few minor comments:

In the description of the artificial infection procedure it is not clear if the blood in the receiver capsule is also supplemented with adenosine triphosphate.

Each source and receiver capsule was supplemented with ATP to improve mosquito feeding. As the information was missing in the manuscript, we added it in the text (lines 85 and 88).

Reviewer 2 Report

The manuscript by Antoine Boullis et al described Zika virus mechanical transmission by Aedes aegypti. In this manuscript, they concluded that the mechanical transmission does not have an impact on ZIKV infection spread, because only a few infectious particles were detected in the mouthparts of partially engorged mosquito. To date, the distinction of biological transmission and mechanical transmission is not clear in zikavirus, therefore, this kind of works is important to understand the Zika epidemiology. Overall, although the experiments were well demonstrated by simple assay, the conclusions are not well supported by the current data. It would be appropriate that the authors provide some additional data for the publication to this journal.

1. The authors used RT-qPCR for viral RNA detection. However, the representation of the RT-PCR results is not suitable. Adequate standards should be used and the absolute value of the RNA copy number should be presented. 

2. How much amount of viruses will be detected in mosquito kept in further period incubation? To evaluate the importance of mechanical transmission accurately, it is also important to check the biological transmission. Assessment how much amount of viruses will be increased in the mosquito is important to know the potential biological transmission in their experimental system. Otherwise, it is hard to image that the numbers indicated in their experiments has some meanings. 

Minor points

1. It would be very helpful for readers to illustrate where a mouthpart and an abdomen correspond with the mosquito's body. 

2. Figure 1 is missing the alphabetical indications.

Author Response

Response to Reviewers

Title: Experimental assessment of Zika virus mechanical transmission by Aedes aegypti

Journal: Viruses

On behalf of all co-authors, I thank the reviewers for their constructive remarks that contributed to improve the quality of our manuscript.

The manuscript was amended following the reviewer’s instructions, with changes highlighted in yellow. In this document, our answers to the reviewers’ comments are highlighted in bold. Line numbers referenced in our comments correspond to the revised version of the manuscript.

Anubis VEGA-RUA (Corresponding author)

Reviewer: 2

Comments and Suggestions for Authors

The manuscript by Antoine Boullis et al described Zika virus mechanical transmission by Aedes aegypti. In this manuscript, they concluded that the mechanical transmission does not have an impact on ZIKV infection spread, because only a few infectious particles were detected in the mouthparts of partially engorged mosquito. To date, the distinction of biological transmission and mechanical transmission is not clear in zikavirus, therefore, this kind of works is important to understand the Zika epidemiology. Overall, although the experiments were well demonstrated by simple assay, the conclusions are not well supported by the current data. It would be appropriate that the authors provide some additional data for the publication to this journal.

1. The authors used RT-qPCR for viral RNA detection. However, the representation of the RT-PCR results is not suitable. Adequate standards should be used and the absolute value of the RNA copy number should be presented.

We agree with your remark. As recommended, we tried to estimate the absolute quantity of viral RNA copies found in the two positive samples. To do so, we used a calibration curve obtained upon ten-fold serial dilutions of standards of known vRNA copies (Altona - Real Star® Zika virus RT PCR kit). However, the CTs obtained for both of our positive samples were high (i.e. 37.9 and 41.9) and under the threshold of quantification. Based on this calibration curve, we can affirm that the PCR mixes contained less than 7 copies of vRNA, meaning that the quantity of viral RNA in each whole sample was less than 500 vRNA copies. These findings are in accordance with those found for the mouthparts of partially engorged mosquitoes, where only 2 viral particles were detected for only one female.

Also, we can confirm that amplification obtained during PCRs corresponded to Zika virus, as the TM obtained for both membrane samples matched with those of the positive controls we used (77.57 and 77.62 °C for both samples, compared to 77.87 and 78.06 °C for the corresponding positive controls).

In the manuscript, we added a figure of the amplification plot for both positive membrane samples (along with their positive controls), with the corresponding TMs and CTs (Figure 2 – lines 170 - 171). In parallel, we added information regarding viral RNA quantification in the methods (lines 136 - 140) and results and discussion sections (lines 180 - 185).

2. How much amount of viruses will be detected in mosquito kept in further period incubation? To evaluate the importance of mechanical transmission accurately, it is also important to check the biological transmission. Assessment how much amount of viruses will be increased in the mosquito is important to know the potential biological transmission in their experimental system. Otherwise, it is hard to image that the numbers indicated in their experiments has some meanings. 

Thank you for the very interesting question and remark. If we reformulate, your question would be the following: what is the biological transmission ability of partially engorged mosquitoes that had an interrupted blood meal? This interesting information is still lacking in the literature and will be useful to better understand vectorial capacity of Ae. aegypti toward Zika virus.

However, our study does not focus on the effect of Ae. aegypti feeding behavior on the biological transmission of Zika but on mechanical transmission. Our goal was to determine experimentally the ability (or not) for Ae. aegypti to transmit mechanically the Zika Virus from an “infected host” to another, without extrinsic incubation period and with an artificial device. We are now aware that this mosquito species is competent for Zika under a biological transmission prospect. We recently published an article about this using the same feeding device were we found high infection, dissemination and transmission rates of the same ZIKV strain we used in this study (Hery et al. 2019, Emerg Microbe Infect, 8: 699-706). Also, it has recently been proven that the colonized Ae. aegypti strain Rockefeller (that we used for mechanical transmission experiments) is competent for ZIKV (Costa Da Silva et al. 2017, PLoS One, 12: e0174081). According to all these previous findings, we decided to only focus on mechanical transmission in the present study.

Minor points

1. It would be very helpful for readers to illustrate where a mouthpart and an abdomen correspond with the mosquito's body. 

The Figure 1 was adapted to well identify the mouthparts of mosquitoes and their abdomen.

2. Figure 1 is missing the alphabetical indications.

We think it was a mistake in edition process. Editors removed the real Figure 1 and used the Graphical Abstract instead. Because the Figure 1 is different from the graphical abstract, we re-added the real figure 1 in the main text.

Round 2

Reviewer 2 Report

The revised manuscript by Antoine Boullis et al seems to be improved compare to the original version, and it is suitable for the publication in this journal.